# Acid Sphingomyelinase Deficiency Type B Patient-Derived Liver Organoids Reveals Altered Lysosomal Gene Expression and Lipid Homeostasis

**DOI:** 10.3390/ijms241612645

**Published:** 2023-08-10

**Authors:** Gema Gomez-Mariano, Sara Perez-Luz, Sheila Ramos-Del Saz, Nerea Matamala, Esther Hernandez-SanMiguel, Marta Fernandez-Prieto, Sara Gil-Martin, Iago Justo, Alberto Marcacuzco, Beatriz Martinez-Delgado

**Affiliations:** 1Molecular Genetics and Genetic Diagnostic Units, Institute of Rare Diseases Research (IIER), Spanish National Institute of Health Carlos III (ISCIII), 28220 Madrid, Spain; sara.perez@isciii.es (S.P.-L.); sheila.ramos@isciii.es (S.R.-D.S.); nmatamala@isciii.es (N.M.); esther.hernandez@isciii.es (E.H.-S.); marta.fernandez@isciii.es (M.F.-P.); sara.gil@externos.isciii.es (S.G.-M.); bmartinezd@isciii.es (B.M.-D.); 2CIBER de Enfermedades Raras, CIBERER U758, 28029 Madrid, Spain; 3General and Digestive Surgery Department, Hospital 12 de Octubre, 28041 Madrid, Spain; iagojusto@hotmail.com (I.J.); alejandro_mar@icloud.com (A.M.)

**Keywords:** acid sphignoimylinase deficiency (ASMD), Niemann–Pick type B, organoids, liver, lipids, lysosome, *SMPD1* gene

## Abstract

Acid sphingomyelinase deficiency (ASMD) or Niemann–Pick disease type A (NPA), type B (NPB) and type A/B (NPA/B), is a rare lysosomal storage disease characterized by progressive accumulation of sphingomyelin (SM) in the liver, lungs, bone marrow and, in severe cases, neurons. A disease model was established by generating liver organoids from a NPB patient carrying the p.Arg610del variant in the *SMPD1* gene. Liver organoids were characterized by transcriptomic and lipidomic analysis. We observed altered lipid homeostasis in the patient-derived organoids showing the predictable increase in sphingomyelin (SM), together with cholesterol esters (CE) and triacylglycerides (TAG), and a reduction in phosphatidylcholine (PC) and cardiolipins (CL). Analysis of lysosomal gene expression pointed to 24 downregulated genes, including *SMPD1*, and 26 upregulated genes that reflect the lysosomal stress typical of the disease. Altered genes revealed reduced expression of enzymes that could be involved in the accumulation in the hepatocytes of sphyngoglycolipids and glycoproteins, as well as upregulated genes coding for different glycosidases and cathepsins. Lipidic and transcriptome changes support the use of hepatic organoids as ideal models for ASMD investigation.

## 1. Introduction

Acid sphingomyelinase deficiency (ASMD) or Niemann–Pick disease (NPD) is a rare lysosomal deposition disease that presents a significant clinical variety depending on the levels of residual acid sphingomyelinase (ASM) activity. Different types are described, NP type A (NPA: OMIM# 257200), NP type B (NPB: OMIM no. 607616), and an intermediate A/B form. The NPA type is the neuronopathic form, with a more severe clinical picture than the NPB type, resulting in the death of patients before 2–3 years [1,2,3,4]. It relates to central nervous system involvement, hepatosplenomegaly, pulmonary disease, and rapid and progressive psychomotor deterioration. The NPB type is the non-neuronopathic clinical form with the manifestation of hepatosplenomegaly, noticeable from infancy, and pulmonary manifestation leading to hepatic and pulmonary failure. In addition, there is an intermediate clinical form A/B characterized by the symptoms of the NPB and slow progressive neurological degeneration of the NPA type [5,6].

This lysosomal disease is caused by biallelic mutations in the sphingomyelin phosphodiesterase-1 (*SMPD1*) gene, located on chromosome 11 (11p15.1–11p15.4), is approximately 5 kb in size and encoding for the ASM enzyme [7,8]. This results in decreased enzyme activity and secondary accumulation of sphingomyelin (SM) and other sphingolipids that become toxic at elevated, non-physiological levels in various tissues. To date, more than 180 mutations have been described within the gene causing NPD types A, B, and A/B, among which there are no hot spots, typically being point mutations (nonsense and nonsense-mediated), small deletions, and splicing abnormalities [9,10,11]. Sphingomyelin (SM) present in the plasma membrane is derived from vesicular transport from the Golgi apparatus. Acid sphingomyelinase on the outer side and neutral sphingomyelinase on the inner side metabolize SM to ceramide (Cer) and other lipids such as sphingosine (Sph). Through endocytic vesicles, the membrane SM reaches the lysosome, where it is degraded to ceramide by acid sphingomyelinase (ASM). In ASMD patients, ASM enzyme deficiency results in SM deposits in the lysosome [12] (Figure 1).

Some of the clinical features that NPB patients present are as follows: 1. Hepatic involvement that can be severe and evolve into chronic hepatitis and fibrosis; sometimes, there are cases of fulminant hepatitis in adulthood. Some patients present portal hypertension or elevation of transaminases and bilirubin [13]; 2. Hemograms of patients may show decreased platelets, hemoglobin, and leukocytes, many of them with thrombocytopenia, which causes bleeding episodes, petechiae, and ecchymosis [14]; 3. Pulmonary involvement due to the accumulation of lipid-laden macrophages in the alveolar septa of the bronchial walls and pleura. Pulmonary lavage is used to prevent lipid deposition and is not very effective; 4. They may present growth problems with delayed skeletal maturation (low weight and height), presenting in some cases with osteopenia and osteoporosis. More than 90% of adult patients with NPB present osteoporosis in one or more locations, leading to bone fractures in at least 58% of cases before the age of 33 years [15]; 5. About 30% of type B patients present with cherry-red eye spots due to lipid accumulation in the retinal ganglion cells [16]. They do not usually present with neurological abnormalities, although, by the end of early childhood, some may present with varying degrees of central nervous system involvement; 6. Hyperlipidemia: They present low serum high-density lipoprotein cholesterol (HDL-C) and high low-density lipoprotein cholesterol (LDL-C) levels accompanied by elevated triacylglycerides (TAG) [17]. Lipid abnormalities are evident from an early age and contribute to the risk of heart disease; 7. Cardiologically, they may present sinus bradycardia, ventricular hypertrophy, and conduction disturbances. Echocardiograms may detect mitral and aortic insufficiency and pulmonary hypertension [5].

Pathological accumulation of SM in the liver results in cirrhosis and liver failure that is also accompanied by abnormal liver enzymes. Since the liver is an important organ involved in lipoprotein biogenesis, lipid metabolism, and homeostasis, it is likely that massive hepatocellular accumulation of SM alters the normal cellular function and leads to alterations in lipid homeostasis [18]. In this work, we established a model of liver disease by generating liver organoids from samples of a Niemann–Pick type B patient, with the aim of characterizing the disease by using omics techniques such as lipidomics to analyze lipid content and transcriptomics to study variations in gene expression.

## 2. Results

### 2.1. Generation of Liver Organoids from a Niemann–Pick Type B Patient (ASMD Type B)

Liver organoid lines were established from a patient with ASMD type B and from a control. From liver biopsies of both, human liver ductal stem cells were isolated, and an undifferentiated (expanding) organoid line was established, which was frozen and characterized following the protocol previously established by our research group [19]. Subsequently and following the protocol, the expanding organoids were cultured for 15 days with a differentiation medium for their differentiation to hepatocytes. Figure 2 illustrates the expansion and differentiation stages of hepatic organoids.

### 2.2. Expression of Differentiation Markers in Liver Organoids

As expected, expanding organoids control and ASMD type B expressed progenitor cell (*LGR5*) and ductal cell (*KRT19*) markers when cultured in an expansion medium. Progenitor cell markers were no longer expressed when organoids were differentiated into hepatocytes. Differentiated organoids showed expression of hepatocyte-specific markers such as albumin (*ALB*) and apolipoprotein B (*APOB*). In addition, the expression of these markers was observed in HepG2 cells and a control liver sample. It should be noted that a much higher expression of *APOB* and *ALB* markers was observed in differentiated ASMD type B organoids compared to control organoids (Figure 3).

### 2.3. The NPB Patient Has the Pathogenic p.Arg610del Variant

Genotyping of the patient revealed a homozygous deletion of an arginine at position 610 of the *SMPD1* gene (GRCh38: NC_000011.10:g.6394540_6394542del, NM_000543.5: c.1829_1831delGCC, NP_000534.3: p.Arg610del). This mutation has been frequently described worldwide and is clearly associated with an attenuated phenotype of ASMD type B disease.

### 2.4. Altered Expression of Sphingomyelin Metabolism Genes

The analysis of the expression of genes involved in sphingomyelin metabolism in ASMD type B versus control organoids showed a significant reduction in ASMD type B organoids in the expression of the *SMPD1* gene coding for the enzyme acid sphingomyelinase (ASM), responsible for the degradation of sphingomyelin to ceramide and phosphatidylcholine. On the other hand, a significantly increased expression of the *SMPD2* gene coding for one of the neutral sphingomyelinases (nSM2) and the *SGMS1* gene coding for one of the sphingomyelin synthases (SMS1) that catalyzes sphingomyelin production was detected (Figure 4).

### 2.5. Increased Intracellular Lipid Content in ASMD Type B Organoids

Oil-red-O staining of ASMD type B and control organoids revealed a significant increase in the number of positively stained cells. In these organoids, hepatocytes exhibited two lipid staining patterns, one showing lipid droplets mostly located close to the cell membrane (Figure 5a) and the other showing a high number of lipid droplets in the cytoplasm (Figure 5d). Globally, the ASMD organoids showed 40% of cells with lipid accumulation, while only 5% of the cells in the control organoids had positive lipid staining, indicating a significant alteration in lipid homeostasis (Figure 5).

### 2.6. Identification of Foam Cells in ASMD Type B Liver Organoids 

ASMD type B organoids also showed histological features typically found in the patients. PAS staining assays of liver organoids from the NPB patient identified the presence of foamy liver cells characteristic of these patients (Figure 5). The cells are referred to as “foam cells” because of their foamy or soap-scum appearance. Lipid-filled foam cells are typically detected in the liver, spleen, lung, bone marrow, and lymph nodes of types A and B. The cells have this foamy appearance due to lipid deposits [1].

### 2.7. Alteration of Lipid Homeostasis in the ASMD Type B Liver Organoids 

Alteration in lipid homeostasis was observed in the ASMD type B liver organoids. Lipidomic analysis of 24 lipids, including phospholipids, glycerolipids, sphingolipids, and sterols, revealed a change in lipid component distribution with a significant increase in some lipid species (Appendix A). Specifically, significantly elevated levels of sphingomyelin (SM), ceramide (Cer) and phosphatidylglycerol (PG), and lysophosphatidylglycerol (LPG), were found, together with some ether-bound lipids, such as phosphatidylcholine (-ether) (PCO-) and lyso-Phosphatidylethanolamine (-ether) (LPEO-). At the same time, cholesterol esters (CE) and triacylglycerols (TAG) also had higher levels in the patient than in the control. On the contrary, reduced levels of phospholipids such as phosphatidylcholine (PC), involved in coagulation processes, and cardiolipins (CL), a major component of the mitochondrial inner membrane where it plays a central role in mitochondrial function, were detected (Figure 6).

### 2.8. Transcriptomic Analysis of Liver Organoids (RNA-Seq)

Data from differential expression analysis of ASMD type B liver organoid versus control showed 4625 altered genes with a *p*-value < 0.01, of which 2255 genes were overexpressed (UP), and 2370 genes were underexpressed (DOWN). Functional annotation of differentially expressed genes using DAVID revealed altered expression of an important proportion of genes within the Lysosome KEEG pathway with 24 under-expressed genes, including the *SMPD1* gene, and 26 over-expressed genes (Appendix A). 

Figure 7 shows a heatmap with differentially expressed lysosomal genes in liver organoids from a Niemann–Pick patient versus control. A significantly lower expression of the *SMPD1* gene, responsible for sphingomyelin degradation, and the prosaponin (PSAP) gene that also degrades sphingolipids in the lysosome was detected. In addition, the expression of genes responsible for vesicular transport from the membrane or from the Golgi to the lysosomes (*AP1G2* and *AP3B1*) and genes encoding lysosomal enzymes with hyaluronidase activity (*HYAL1* and *HYAL2*) was also reduced. Moreover, a gene involved in the maintenance of lipid homeostasis (*LIPA*) and another responsible for platelet activation (*CD63*) also showed downregulated expression. On the other hand, an altered expression of genes that have been associated with other lysosomal deposition diseases such as Fucosidosis, Cystinosis, Metachromatic Leukodystrophy, Morquio A Syndrome, Gangliosidosis type AB or the AB variant of Tay-Sachs disease and Sandhoff disease was detected. Likewise, genes overexpressed in ASMD l type B liver organoids identified glycosidases, cathepsins (B, D, L1, and S) involved in protein degradation, genes involved in organelle acidification (*TCIRG1* and *ATP6V0B*), ceramidases (*ASAH1*), and genes encoding for proteins responsible for cholesterol transport such as *NPC1* and *SORT1*.

## 3. Discussion

Acid sphingomyelinase deficiency (ASMD) or Niemann–Pick type A, B, and A/B is an autosomal recessive disease with complex clinical features [20,21]. The lysosomal deposition of sphingomyelin due to the lack of functional ASM enzyme results in the involvement of many organs, the liver being the one that gives rise to hepatomegaly in patients. Studying liver involvement from patient samples is limited, mainly because it is a rare disease, but on the other hand, because of the difficulty of maintaining cultures of primary human hepatocytes. Currently, the establishment of 3D organoid cultures is being used as a model for the study of diseases [19,22,23]. The establishment of a liver organoid culture from a Niemann–Pick type B patient and a healthy control allowed us to characterize key features of the liver disease.

The patient included in this study has the Arg610del mutation in the *SMPD1* gene described by Levran and collaborators in 1991, associated with an attenuated phenotype of NPB [24]. The frequency and distribution of *SMPD1* gene variants differ among different populations and ethnic groups, Arg610del being the most frequently reported mutation. Several studies of the frequency and distribution of variants in the *SMPD1* gene in southern European populations have revealed that this variant constitutes 100% of the cases of Niemann–Pick Type B in the Canary Islands [25] and 61.5% in Spain, being the most frequent mutation [26]. This variant is classified in ClinVar as pathogenic (ID: VCV000198093.27). Studies have been published showing an impact on protein function due to impaired proteolytic maturation of the enzyme, associated with a reduction in enzyme activity. However, this variant is associated with the less severe type B phenotypes as it is able to maintain a residual activity close to 21.5% compared to the activity of a wild-type enzyme [26].

ASMD type B organoids revealed altered expression of genes directly involved in sphingomyelin metabolism. On the one hand, decreased expression of the *SMPD1* gene would result in increased SM deposits in the cell characteristic of patients with NP disease and, on the other hand, increased expression of the *SMPD2* gene, which is likely to be a cellular mechanism to regulate excess SM. In addition to the reduced expression of the *SMPD1* gene, increased expression of the *SGMS1* gene, which synthesizes SM from phostidylcholine and ceramide, was observed, which would also be contributing to the increased SM deposits. Although the mechanism of *SGMS1* overexpression is not known, this change in gene expression could be a response to the significantly elevated ceramide levels found by the lipidomics study in ASMD type B organoids.

Transcriptomic analysis revealed a significantly reduced expression of 24 lysosomal genes in ASMD type B liver organoids compared to the control. As expected, a lower expression of the *SMPD1* gene, responsible for sphingomyelin degradation, as well as of a prosaponin (*PSAP*) [27] that also degrades sphingolipids in the lysosome, was detected. PSAP is a highly conserved preprotein that is proteolytically processed to generate saposins A, B, C, and D that catabolize sphingoglycolipids in the lysosomal compartment Mutations in this gene have been associated with Gaucher disease [28] and Metachromatic leukodystrophy [29]. A reduction in the expression of both genes could lead to the accumulation of sphingophospholipids (sphingomyelin) and sphingoglycolipids in the ASMD type B liver organoids. In addition to reduced expression of these genes, significant downregulation of *FUCA1*, *CTNS*, *ARSA*, *GALNS*, *GM2A*, and *HEXB* genes could also contribute to the accumulation in the lysosome of sphingoglycolipids such as gangliosides and cerebrosides, glycosaminoglycans, fucose-containing glycolipids, and glycoproteins. Mutations in these genes are associated with other lysosomal storage diseases [30,31,32], such as Fucosidosis, Cystinosis, Metachromatic leukodystrophy, Morquio A syndrome, Sandhoff disease, Gangliosidosis type AB or AB variant of Tay–Sachs disease and GM2-gangliosidosis type II, respectively. Additionally, ASMD type B organoids showed overexpression of genes encoding enzymes that catalyze the hydrolysis of glycosidic bonds, such as *GAA,* which degrades glycogen to glucose, *IDUA* and *NAGA* enzymes that degrade glycosaminoglycans and glycosyl hydrolases such as *MAN2B1* and *MANBA* [11,33]. Consequently, it is likely that the cell could overexpress these glycosidases to regulate sphingoglycolipid deposits generated by the reduced expression of enzymes that catalyze their degradation. 

Furthermore, to these deposits of sphingolipids and sphingoglycolipids, it is possible that there is an accumulation of lysosomal proteins since the altered expression of several cathepsin genes was detected in the ASMD type B organoids. Cathepsins are responsible for the hydrolysis of proteins [34,35,36,37]. On the one hand, decreased expression of cathepsins A, E, F, H, V, and Z was observed. Interestingly, mutations in cathepsin A generate significant protein deposits in the cell associated with galactosialidosis [38]. On the other hand, an increase in the expression of cathepsins B, D, L1, and S was identified. The overexpression of cathepsin B has also been described before causing liver fibrosis in patients with Niemann–Pick disease [39]. In addition, ASMD type B organoids showed high expression levels of *LAMP2*, a lysosomal membrane glycoprotein that participates in protein degradation playing an important role in chaperone-mediated autophagy. Some authors point out that lysosomal deposition diseases all converge in a disorder of the normal function of the lysosome during the autophagy process. Lysosomal deposits due to a lack of functional proteins result in collateral damage consisting of a secondary accumulation of autophagy debris [40,41].

Increased expression of genes such as *TCIRG1* and *ATP6V0B*, which are involved in the maintenance of organelle acidification, regulating the pH of cells and their environment by proton pumping (V-ATPase) were also detected in ASMD type B organoids. This acidification is required for intracellular processes such as protein sorting, zymogen activation, and receptor-mediated endocytosis [42].

Another overexpressed gene was the lysosomal acid ceramidase *ASAH1* (N-acylsphingosine amidohydrolase1) that hydrolyzes ceramides to sphingosine and free fatty acids at acidic pH. Significantly, higher levels of ceramide were detected in the ASMD type B organoids compared to the control. Ceramides are bioactive lipids that mediate cell signaling proliferation, apoptosis, and differentiation [43]. Overexpression of the *ASAH1* gene could be a mechanism to regulate elevated ceramide levels. Ceramide is a product of the hydrolysis of sphingomyelin by ASM, which is defective in ASMD type B organoids. Therefore, elevated levels of ceramide could be due to the action of other sphingomyelinases present in non-lysosomal compartments, as has been described by other authors in ASMKO mice [1].

LIPA belongs to the lipase family and is crucial for the intracellular hydrolysis of cholesterol esters (CE) and triacylglycerols (TAG) in lysosomes. A reduced expression of the *LIPA* gene in the organoids is consistent with the elevated levels of CE and TAG detected in the lipidomics study. However, genes of lysosomal membrane proteins such as *SLC17A5*, *SORT1*, *LAMP2*, *NPC1*, and *SCARB2*, involved in cholesterol and protein transport, showed significantly higher expression in ASMD type B organoids. *SORT1* codes for sortilin, an alternative receptor to mannose-6-phosphate, which is involved in the trafficking of ASM to lysosomes [44]. Increasing the expression of *SORT1* would favor the transport of ASM to the lysosome to degrade the SM deposits.

The most highly altered lysosomal genes in ASMD type B organoids were the tartrate-resistant acid phosphatase (*ACP5* or *TRAP*) involved in bone resorption processes and the Cathepsin V (*CTSV*), a cysteine transporter outside the lysosome. Skeletal pathologies are very frequent in lysosomal deposit diseases [45]. One of the clinical characteristics of Niemann–Pick type B patients is osteoporosis. Transgenic mice overexpressing *TRAP* have increased osteoclastic resorption of bone. Its expression seems to increase in certain pathological states such as Gaucher and other diseases. In bone resorption processes, the levels of TRAP increase in the serum of patients and are therefore routinely used to monitor responses to treatment in Gaucher disease [46]. Sharma et al. described two Niemann–Pick type B patients who had elevated TRAP levels in bone marrow foam cells [47]. The other more significantly altered gene, *CTSV* [48], a transporter located in the lysosomal membrane that carries cysteine outside of the lysosome, had a highly reduced expression in ASMD type B organoids, which could contribute to increased lysosomal stress by cysteine deposition.

Although all these gene expression changes focused on lysosomal genes, allowing a better understanding of the disease, there were a substantial number of other differentially expressed genes found in the ASMD type B liver organoid model to be further analyzed in future studies.

The lipidomic analysis revealed alterations of specific lipid species in ASMD type B organoids. As expected, significantly higher sphingomyelin (SM) levels, of a 1.6-fold increase, were detected in ASMD type B liver organoids. In addition, triacylglycerols (TAG), which are a source of energy being a reservoir of structural and bioactive fatty acids, showed twofold increased levels in ASMD type B organoids with respect to the control. The accumulation of TAG in tissues is the origin of health problems such as diabetes, fatty liver, and cardiovascular diseases. Early coronary heart disease has been described in ASMD patients and has been related to dyslipidemia, an alteration in blood lipid levels, mainly cholesterol and triglycerides [5]. In addition, lower DAG levels were observed in the ASMD type B organoids, which is a precursor of TAG. In this study, we also detected elevated cholesterol ester (CE) levels in the patient organoids that could also be contributing to cardiovascular disease associated with ASMD. Patients would accumulate cholesterol in the form of cholesterol esters in atherosclerotic plaques, leading to cardiovascular disease. However, it is still difficult to establish a direct relationship between elevated sphingomyelin and cardiovascular problems in patients with ASMD. Nevertheless, it a correlation between the accumulation of SM and the alteration of lipid homeostasis has been established [1].

ASMD patients typically present coagulation problems because of a low number of platelets. In this study, lower levels of phospholipid involved in the coagulation process, phostatidylcholine (PC), a component of the platelet-activating factor, were detected in ASMD type B organoids, which could relate to the coagulation problems associated with the disease. Furthermore, a significant reduction in the expression of the *CD63* gene, a marker of blood platelet activation [49], was found in the ASMD type B liver organoids. The lack of CD63 expression has also been described in patients with Hermansky–Pudlak syndrome (HPS), which has hemorrhages due to platelet deficiency, among other symptoms [50].

ASMD patients have lipid deposits in the retina that give rise to a cherry-red spot [1]. A significant increase of ether-bound lysophosphatidylethalonamines (LPEO-) was detected in the ASMD type B liver organoids. In retinal epithelial cells, ether-bound lysophosphatidylethalonamines (LPEO-) are precursors of bisretinoids, a family of fluorescent molecules formed in photoreceptor cells [51]. Bisretinoid deposition in the retinal pigment epithelium leads to some retinal diseases, such as Stargardt’s disease (STGD), which manifests as retinal blotches around the macula. These results suggest studying LPEO- levels in retinal epithelial cells because they could contribute to the ocular pathology found in ASMD patients.

Cardiolipins (CL) are major components of the mitochondrial inner membrane (IMM) along with other phospholipids. It has been described by other authors that changes in the phospholipid composition of the mitochondrial membrane directly affect integrity, fluidity, and membrane permeability, in turn altering the activity of IMM proteins [52]. In this study, lower levels of three of these phospholipids, CL, PE, and PC, were detected in ASMD type B organoids, which could relate to the impaired mitochondrial function described in patients with Niemann–Pick type B [53,54,55]. Defects in cardiolipin content could have implications for the regulation of mitochondrial function. Alterations in cardiolipins have been linked to the pathogenesis and progression of various metabolic diseases, cardiomyopathies, Barth’s syndrome, and Parkinson’s disease [56,57].

In summary, our results show that ASMD type B patient-derived liver organoids are suitable models for investigating molecular alterations of liver disease. However, it should be taken into account that results reflect alterations found in specific type B ASMD patients and control individuals. The development of organoids from additional patients with other mutations and the generation of isogenic controls might be very useful to confirm common altered processes and expand our knowledge of liver disease in ASMD patients. 

## 4. Materials and Methods 

### 4.1. Patients

Liver organoids were established from the biopsies of two patients from Hospital 12 de Octubre in Madrid (Madrid, Spain). The first was a control patient from whom a liver biopsy was obtained from the healthy part of the liver by surgical intervention due to hepatocellular carcinoma. The other corresponds to a patient with Niemann–Pick disease type B who presented hepatosplenic involvement and pulmonary emphysema, with an ASM enzymatic activity of 15%. Informed consent was signed by the patients to be included in this study and approved by the ethics committee of the Institute of Health Carlos III of Madrid (Madrid, Spain). The study was conducted in accordance with the Declaration of Helsinki and approved by the Ethics Committee of the Institute of Health Carlos III (protocol code CEI PI 27_2021-v3).

### 4.2. Genotyping

Control and Niemann–Pick patients were genotyped for *SMPD1* by DNA extraction from a liver biopsy with the NucleoSpin Tissue de MACHEREY-NAGEL kit. The six exons of the gene and their flanking regions were studied by PCR with the primers shown in Appendix A and designed by our group. They were then sequenced in forward and reverse with the automatic sequencing device (ABI PRIMS 377 Applies BioSystems, Foster City, CA, USA). Sequencing was performed using previously described primers in an automatic sequencer (ABI PRISM 377 Applied BioSystems). 

Sequence analysis revealed a homozygous mutation in exon 6 consisting of a deletion of the amino acid arginine at position 610 (NM_000543.5 (*SMPD1*): c.1829_1831del; NP_000534.3: p.Arg610del). This mutation results in an in-frame deletion that was predicted to remove an arginine amino acid in the protein. The identified mutation was confirmed in a duplicate PCR product.

### 4.3. Establishment and Culture of Human Liver Organoids

Liver organoids were generated from biopsies of a Niemann–Pick type B patient and a control, with the aim of establishing a model for the study of liver disease. The protocol described by Gomez-Mariano et al. [19,58] was followed. Organoids were generated from adult human liver progenitor cells. After surgical excision, the tissue was stored in a basal medium (advanced DMEM/F12, 1% penicillin/streptomycin, 1% Glutamax, 10 mM Hepes) for 24 h, then treated with collagenase to disaggregate (Earle’s equilibration saline solution (EBSS), collagenase D 2.5 mg/mL, DNAse I 0.1 mg/mL). After centrifugation, cells were enclosed in a BME2 matrix (Cultrex Basement Membrane Extract BME Type 2, BME2) and cultured with an isolation medium (basal medium supplemented with 25 ng/mL recombinant human Nogging, 30% conditioned medium with Wnt3a, and 10 µM Y-27632 ROCK (Rho-associated, coiled-coil containing protein kinase) inhibitor). In this specific culture condition, only progenitor AdSCs, located mainly in the bile ducts, can grow, forming organoids with self-renewal capacity in the form of spheres. For culture expansion, an expansion medium (basal medium, 1 mM N-acetylcysteine, 5% Rspo1 conditioned medium, 10 mM nicotinamide, 10 nM recombinant human-gastrin I, 50 ng/L recombinant EGF, 100 ng/mL recombinant human FGF10, 50 ng/mL recombinant human HGF, 10 µM ROCK inhibitor) was used. For differentiation to hepatocytes, they were cultured with a differentiation medium (basal medium, 1:50 B27, 1:100 N2 supplement, 1 mM N-acetylcysteine, 10 nM recombinant human gastrin I, 50 ng/mL recombinant mouse EGF, 100 ng/mL recombinant human FGF10, 50 nM A83-01, 10 µM DAPT). Figure 2 illustrates the stages of organoid development: isolation, expansion, and differentiation.

### 4.4. Validation of Progenitor, Ductal, and Differentiation Cell Markers

Validation of specific expression markers was performed by RT-qPCR on RNA from organoids (expanded and differentiated), HepG2 cells, and liver biopsy. RNA extraction was performed with RNeasy Mini (Qiagen, Hilden, Germany) and cDNA synthesis with the Maxima First Strand cDNA Synthesis kit (Thermo Scientific, Waltham, MA, USA). The assay qPCR was performed in triplicate using the Taqman Fast Advance master mix (Applied BioSystems^®^, Waltham, MA, USA), specific primers and Taqman probes from the Universal probe library, UPL, (Roche, Mannheim Germany): KRT19 #71), LGR5 (#78), ALB (#44), APOB (#90), (Appendix A) [19]. 

The qPCR was performed on a QuantStudio 5 System (ThermoFisher Scientific, Waltham, MA, USA) and analyzed using the QuantStudio Design and Analysis Software v1.4.3. Glyceraldehyde-3-phosphate dehydrogenase (GADPH) as an endogenous qPCR control, and the relative expression obtained was calculated using the comparative Ct method and obtaining the fold change value (DDCt).

### 4.5. Expression Analysis of Sphingomyelin Metabolism Genes 

We quantitatively analyzed the expression of three sphingomyelin metabolism genes, acid sphingomyelinase (ASM), neutral sphingomyelinase 2 (nSM2), and sphingomyelin synthase (SMS1). Expression was compared by RT-qPCR from organoid RNA differentiated to control and patient NP type B hepatocytes. The SYBRGreen qPCR assay was performed from cDNA (diluted 1/10) obtained from 1 μg RNA by reverse transcription using the Maxima First Strand cDNA Synthesis Kit (Thermo Scientific, Fermentas Life Sciences, St. Leon-Rot, Germany). SYBRGreen qPCR assays were performed in triplicate on a QuantStudio (AppliedBiosystems^®^, Waltham, MA, USA) in 10 µL reaction volume containing 5 µL SYBRGreen PCR Mastermix and 10 nM of each primer was used. Specific primers were used for each gene (Appendix A). The following thermal program was applied: a single cycle of DNA polymerase activation for 20 sec at 95 °C followed by 45 amplification cycles of 1 sec denaturing step (95 °C) and 20 sec annealing–extension step (55 °C). Afterward, melting temperature analysis of the obtained amplification products was performed using standard machine settings. Note that the fluorescent reporter signal was normalized against the internal reference dye (ROX) signal. Relative expression was calculated using the comparative Ct method and obtaining the fold-change value (ΔΔCt). The ΔCt value of each sample was calculated using GAPDH as an endogenous gene control. *p* values below 0.05 were considered statistically significant.

### 4.6. Determination of Intracellular Neutral Lipid

Oil-red (ORO) staining (Sigma Aldrich- MAK 194, Madrid, Spain) was used to determine intracellular neutral lipid accumulation. Differentiated organoid samples prepared on slides were washed with PBS and 60% isopropanol before labeling with oil-red at 37 °C for 5 min, then washed again with 60% isopropanol and PBS. The nuclei were stained with Mayer’s hematoxylin (Panreac AppliChem, Barcelona, Spain) for 5 min and, after removing the excess dye, prepared with Eukitt mounting medium. They were then visualized on a Leica DFC 7000T microscope (Leica Microsystems, Wetzlar, Germany). For lipid quantification, the preparations were scanned with the NanoZoomer-SQ Hamamatsu 2D Scanner and then quantified with Image Pro Plus software v1.48.

### 4.7. Lipid Extraction, Mass Spectrometry (MS) Data Acquisition and Data Analysis and Post-Processing

Lipotype GmbH (Dresden, Germany), as described, performed mass spectrometry-based lipid analysis [59]. Samples were processed with the Lipotype Shotgun Lipidomics platform allowing automated direct sample infusion and high-resolution Orbitrap mass spectrometry, including lipid class-specific internal standards to ensure absolute lipid quantification. The equipment for extraction and sample infusion was Hamilton Robotics STARlet and Advion Triversa Nanomate, respectively. 

Before performing the lipid extraction with chloroform and methanol, the samples were spiked with a lipid standard [60]. Mass spectra were acquired on a hybrid quadrupole/Orbitrap mass spectrometer equipped with an automated nano-flow electrospray ion source in both positive and negative ion mode with a resolution of Rm/z = 200 = 280,000 for MS and Rm/z = 200 = 17,500 for MSMS, in a single acquisition (Thermo Scientific Q-Exactive). Both MS and MSMS data were combined to monitor CE, Chol, DAG, and TAG ions as ammonium adducts; LPC, LPC O-, PC, PC O-, as formiate adducts; and CL, LPS, PA, PE, PE O-, PG, PI, and PS as deprotonated anions. MS only was used to monitor LPA, LPE, LPE O-, LPG, and LPI as deprotonated anions; Cer, HexCer, and SM as formiate adducts and cholesterol as ammonium adduct of an acetylated derivative [61]. After extraction and processing of the samples, the signal-to-noise ratio of the internal standard showed a high quality of the spectra. 

Using –LipotypeXplorer- software [62], lipid identification by mass spectrometry was performed. Data analysis is performed using LipotypeZoom, a web browser-based data visualization tool, and Lipotype LIMS [62]. Only lipids meeting the criterion of intensity greater than five times over the noise and over the blank samples were taken into account for data analysis. Depending on the nature of the raw data, lipid molecules may be identified as species or subspecies; a list of the lipids quantified in each sample studied is shown in Appendix A. The identified lipid molecules were quantified by normalization to a lipid class-specific internal standard. The amounts in pmoles of individual lipid molecules (species of subspecies) of a given lipid class were summed to yield the total amount of the lipid class.

### 4.8. Foam Cell Detection

For detecting foam cells by PAS staining, organoid liver cell cytospins were stained with periodic acid–Schiff. Organoids were disgregated and incubated in TrypLE Express (GibcoTM, ThermoFisher Scientific) at 37 °C, then washed with phosphate-buffered saline (PBS). The cells were spread on slides by centrifugation for 1 min at 600 rpm and fixed with 4% PFA. They were then washed twice with PBS and stained with hematoxylin–eosin and PAS. On the other hand, paraffin-embedded liver organoid sections of 3 µm were stained with hematoxylin–eosin.

### 4.9. Transcriptomic Analysis of Liver Organoids (RNA-Seq)

Total RNA was isolated from organoid cells using RNeasy Mini (Qiagen, Hilden, Germany), then treated with DNaseI to remove genomic DNA from the samples. RNA purity was assessed using the Agilent RNA 6000 Nano kit and Agilent 2100 Bioanalyzer. Following the manufacturer’s recommendations, the TruSeq Stranded mRNA Kit (Illumina, San Diego, CA, USA) was used. At the Genomics Service (Institute of Health Carlos III), sequencing of the library was performed on a NextSeq 500 sequencer using 75-base read lengths in paired-end mode. The RNA-Seq data obtained were analyzed by the Bioinformatics Service (Institute of Health Carlos III) with an RNA-seq pipeline (https://github.com/BU-ISCIII/rnaseq-nf, accessed on 20 May 2022) written in Nextflow (https://www.nextflow.io/, accessed on 20 May 2022) based on the ones written by nf-core (https://nf-co.re/, accessed on 20 May 2022) (https://github.com/nf-core/rnaseq, accessed on 20 May 2022) GEO accession number: GSE 236213.

Functional characterization of differentially expressed genes (DEGs) was investigated using the Database for Annotation, Visualization, and Integrated Discovery (DAVID Knowledgebase (v2023q1)). Differentially expressed genes (DEGs) were considered when false discovery rate (FDR) values were <0.01 [63,64].

## 5. Conclusions

In this study, lipidomic and transcriptomic analysis revealed alteration of lipid content homeostasis and aberrant expression of lysosomal genes in the ASMD type B liver organoid model. This novel organoid model allowed us to identify and quantify differences in the level of specific lipid species in addition to SM, such as the increase in TAG and CE, likely explaining the alteration in lipid homeostasis or the decrease of PC levels that could be associated with the coagulation problems, generally found in ASMD patients. Moreover, additional accumulation of sphingoglycolipids and possibly proteins were also found to contribute to lysosomal stress. Liver organoid lines from NPB patients with the same and other mutations would help to better understand the mechanisms of disease and to establish unknown phenotype–genotype associations, constituting a very valuable material for testing new treatments as a basis for personalized medicine.

## Figures and Tables

**Figure 1 ijms-24-12645-f001:**
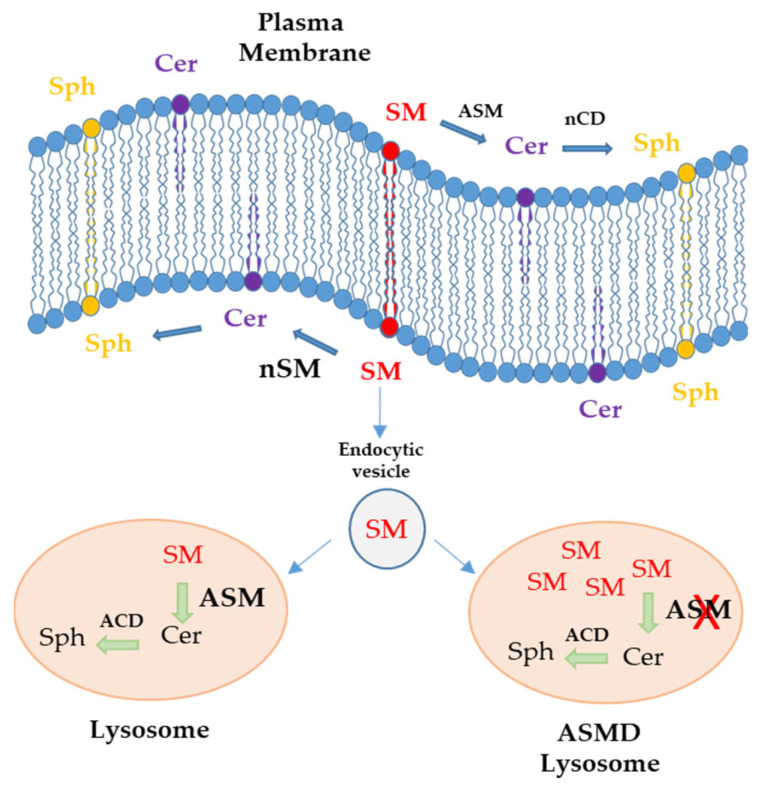
Schematic representation of the localization of sphingomyelin (SM), ceramide (Cer), and sphingosine (Sph) in the plasma membrane and lysosome. Internalization of SM from the plasma membrane through endocytic vesicles to the lysosome leads to the synthesis of ceramide by the action of acid sphingomyelinase (ASM) in healthy individuals. On the other hand, in ASMD patients, SM accumulates in lysosomes due to ASM deficiency; neutral sphingomyelinase (nSM), neutral ceramidase (nCD), and acid ceramidase (ACD).

**Figure 2 ijms-24-12645-f002:**
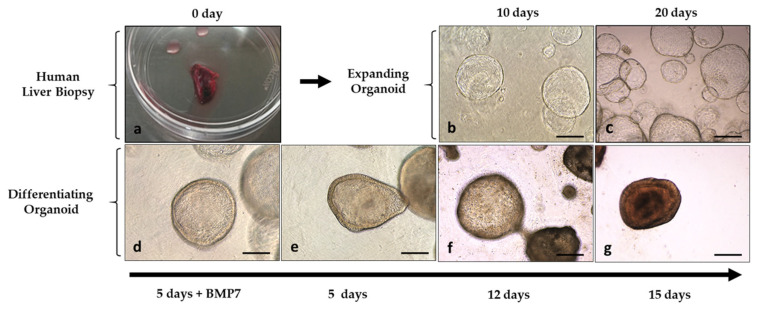
Images of the establishment of an expanding organoid culture from liver biopsies and its differentiation to hepatocytes. (**a**) Human liver biopsy; (**b**,**c**) liver organoids growing with 3D matrigel in expansion medium; (**d**) early days of the differentiation process where organoids grow 5 days in expansion medium with BMP7; (**e**,**f**) differentiating organoids; (**g**) differentiated organoids after 15 days growing in differentiation medium; magnification ×10, Microscope Leica DMIL LED, Leica MC170HD camera; scale bar represents 250 µm.

**Figure 3 ijms-24-12645-f003:**
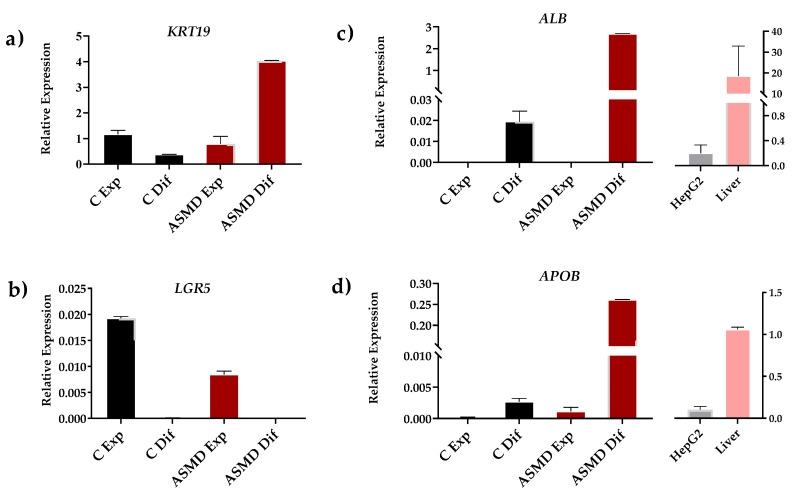
Gene expression analysis of differentiation markers in control (C) (black) and ASMD type B organoids (red) under expansion (Exp) and differentiation (Dif) conditions. (**a**,**b**) Expression of ductal (*KRT19*) and pluripotent (*LGR5*) cell markers; (**c**,**d**) Expression of differentiated hepatocyte markers (*ALB* and *APOB*). These markers were also analyzed in HepG2 cells (gray) and in liver biopsy (pink). Error bar shows standard deviation.

**Figure 4 ijms-24-12645-f004:**
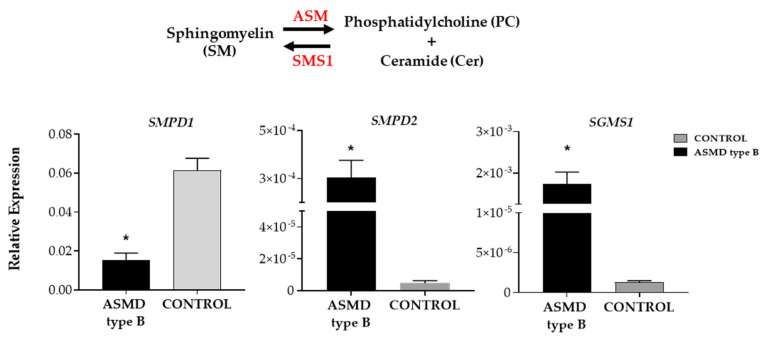
The relative expression of ASMD type B and control liver organoids of three genes of sphingomyelin metabolism. SMPD1 gene coding for ASM (acid sphingomyelins), SMPD2 gene coding for nSM2 (neutral sphingomyelinase 2), and SGMS1 gene coding for SMS1 (sphingomyelin synthase 1) is shown. Statistically significant values are shown by * (*p*-value < 0.05). Error bar shows standard deviation.

**Figure 5 ijms-24-12645-f005:**
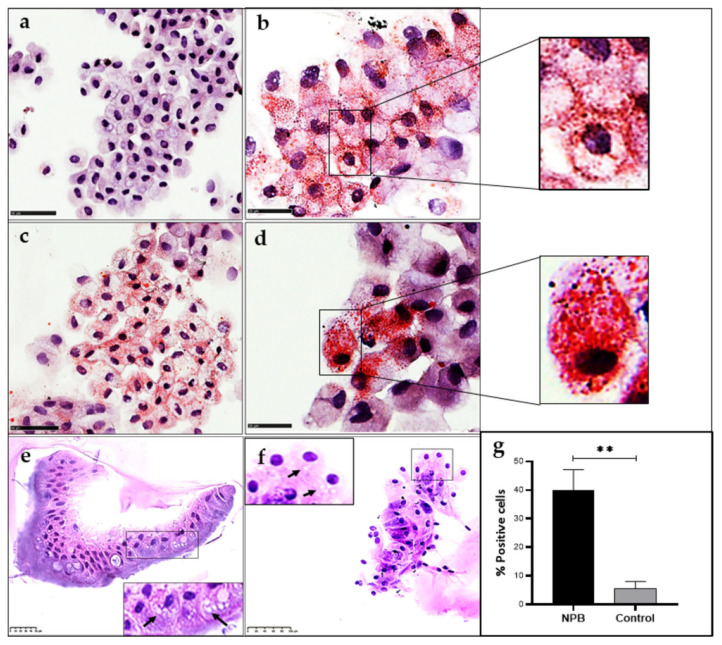
Oil Red-O staining of organoids differentiated to hepatocytes: (**a**) Control organoids; (**b**–**d**) ASMD type B organoids. Cells stained in red show intracellular neutral lipids (Scale 50 µm (**a**–**c**) and 25 µm (**b**–**d**)). Foamy cells are shown in ASMD type B liver organoids stained with hematoxylin-eosin: (**e**) Paraffin-embedded organoid slices (scale 50 µm); (**f**) Cytospin cells from organoids (scale 100 µm). (**g**) The % of cells positive for intracellular neutral lipids are shown in control and ASMD type B organoids after quantification with Image Pro Plus. Statistically significant values are shown by ** (*p*-value < 0.005). Error bar shows standard deviation.

**Figure 6 ijms-24-12645-f006:**
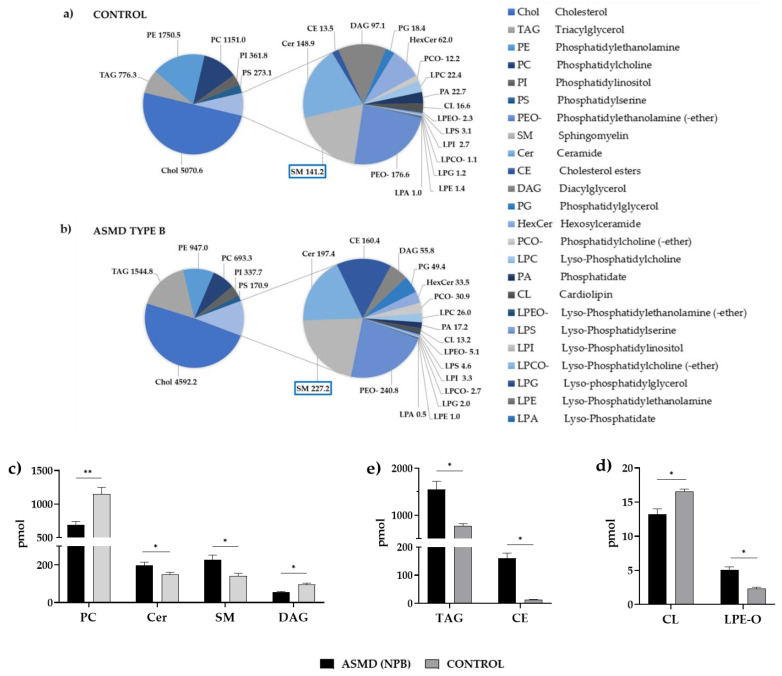
(**a**,**b**) The pie charts show the amount in pmoles of the 24 lipids studied for the control and ASMD type B organoids. (**c**,**d**) Amount of lipids in pmol in the control (gray) and ASMD type B organoids (black): (**c**) Lipids of sphingomyelin metabolism: Ceramide (Cer), sphingomyelin (SM), Diacylglycerol (DAG) and Phosphatidylcholine (PC); (**d**) Triacylglycerol (TAG) and cholesterol esters (CE) (**e**) Cardiolipins (CL) and ether-bound lysophosphatidylethalonamines (LPEO-); * *p*-value < 0.05 and ** *p*-value < 0.005. Error bar shows standard deviation.

**Figure 7 ijms-24-12645-f007:**
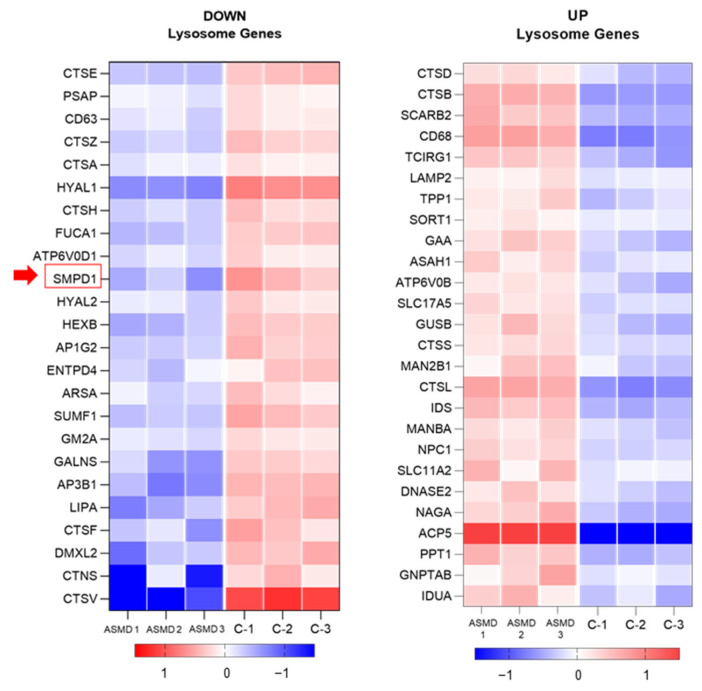
Underexpressed (DOWN) and overexpressed (UP) lysosomal gene. The *SMPD1* gene (red arrow) is shown to be less expressed in ASMD type B liver organoids than in control organoids.

## Data Availability

Data accessible at the NCBI GEO database, accession number: GSE 236213.

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
