# Peer review of "Acid Sphingomyelinase Deficiency Type B Patient-Derived Liver Organoids Reveals Altered Lysosomal Gene Expression and Lipid Homeostasis"

_ijms, 2023, doi:10.3390/ijms241612645_

Round 1

Reviewer 1 Report

In this work, the authors have established liver organoids from a control individual and a patient suffering from the non-neuronophatic form of the acid sphingomyelinase deficiency, also called Niemann Pick type B (NPB). To my knowledge this is the first attempt to generate human liver organoids in NPB, which is characterized by hepatomegaly and liver failure. Therefore, the study is relevant and timely. The manuscript is well written with a nice introduction and discussion. Besides generating the organoids, the authors have performed lipidomic and transcriptomic analysis. The results not only validate the model as a suitable one to investigate NPB but also provide interesting data that helps understanding NPB liver pathology and may open therapeutic perspectives. The lipidomic analysis confirms alterations in sphingolipids. It also unveils imbalance in other lipids such as cholesterol esters, phosphatidylcoline or ether bound- lysophosphatidylethanolamines, which may contribute, respectively,  to cardiovascular disease, coagulation problems, or ocular pathology that characterize NPB beyond liver failure. In addition, the report of alterations in gene expression of lysosomal proteins contribute to understand lysosomal stress in this disease. The manuscript may benefit from addressing the following queries:

1.The control organoid was obtained from the healthy part of the liver biopsy of a patient with hepatocellular carcinoma. Authors may want to discuss the suitability of this control and the pros and cons of not having an isogenic control for the liver organoid derived from the NPB patient

2.The authors performed an appropriate quality check of the liver organoids by analysing the expression of differentiation markers and of hepatocyte-specific markers. It would be convenient to check for the expression of non-liver markers as an additional control.

3.Despite the clear increase in sphingomyelin levels in the NPB liver organoid, the authors find the surprising upregulation of a sphingomyelin-synthetic gene, SGMS1, with respect to the control. Discussion on this unexpected finding might be worth it

4.The authors have analysed the appearance of neutral lipid accumulation and of foam cells in the liver organoids. Are there also differences in the size of the liver organoids derived from the control individual compared to those derived from the NPB patient?

Minor point

Check English language in the legend of Figure 1

Reviewer 2 Report

In this work, the authors established a model of liver disease by generating liver organoids from samples of a Niemann Pick type B patient, with the aim of characterizing the disease by using omics techniques such as lipidomics to analyze lipid content and transcriptomics to study variations in gene expression.

The idea of this study - is interesting; but his manuscript needs some improvements and corrections before publishing may be possible.

General points:

Please add a list of abbreviations before References section to your manuscript.

Special points:

Introduction

Lines 30-42: please add multiple references at the end of each these sentences.

Lines 43-55: please add multiple references at the end of each these sentences.

Lines 62-79: please add multiple references at the end of each these sentences.

 Results

Figure 6 and Figure 7: please provide a Figures with better quality.

Discussion

Lines 213-219: please add multiple references at the end of each these sentences.

Lines 244-256: please add multiple references at the end of each these sentences.

Lines 267-273: please add multiple references at the end of each these sentences.

Lines 320-322: please add multiple references at the end of each these sentences.

Materials and Methods

First of all, please add to this section the exactly name of the organization, the exactly date and the number of the permission of all your experiments.

Please add to each method the appropriate references according to which group or publication you did this method.

Conclusion

Please add Conclusion and Future Perspectives section to your manuscript.

Round 2

Reviewer 2 Report

Thank you for all corrections.